# Malaria and helminth co-infections in children living in endemic countries: A systematic review with meta-analysis

Muhammed O. Afolabi[1]*, Boni M. Ale[2], Edgard D. Dabira[3], Schadrac C. Agbla[4], Amaya L. Bustinduy[5], Jean Louis A. Ndiaye[6,7], Brian Greenwood[1]

**1** Department of Disease Control, London School of Hygiene & Tropical Medicine, London, United Kingdom, **2** Holo Healthcare Limited, Nairobi, Kenya, **3** Disease Control and Elimination Theme, Medical Research Council Unit The Gambia at London School of Hygiene & Tropical Medicine, Fajara, The Gambia, **4** Department of Health Data Science, University of Liverpool, Liverpool, United Kingdom, **5** Department of Clinical Research, London School of Hygiene & Tropical Medicine, London, United Kingdom, **6** Department of Parasitology, University of Thies, Thies, Senegal, **7** Département de Parasitologie-Mycologie, Université Cheikh Anta Diop, Dakar, Senegal

* Muhammed.Afolabi@lshtm.ac.uk

## Abstract

### Background

Current knowledge on the burden of, and interactions between malaria and helminth co-infections, as well as the impact of the dual infections on anaemia, remains inconclusive. We have conducted a systematic review with meta-analysis to update current knowledge as a first step towards developing and deploying coordinated approaches to the control and, ultimately, elimination of malaria-helminth co-infections among children living in endemic countries.

### Methodology/Principal findings

We searched Medline, Embase, Global Health and Web of Science from each database inception until 16 March 2020, for peer-reviewed articles reporting malaria-helminth co-infections in children living in endemic countries. No language restriction was applied. Following removal of duplicates, two reviewers independently screened the studies for eligibility. We used the summary odds ratio (OR) and 95% confidence intervals (CI) as a measure of association (random-effects model). We also performed Chi-square heterogeneity test based on Cochrane's Q and evaluated the severity of heterogeneity using $I^2$ statistics. The included studies were examined for publication bias using a funnel plot and statistical significance was assessed using Egger's test (bias if p<0.1).

Fifty-five of the 3,507 citations screened were eligible, 28 of which had sufficient data for meta-analysis. The 28 studies enrolled 22, 114 children in 13 countries across sub-Saharan Africa, Southeast Asia and South America. Overall, the pooled estimates showed a prevalence of *Plasmodium*-helminth co-infections of 17.7% (95% CI 12.7–23.2%). Summary estimates from 14 studies showed a lower odds of *P. falciparum* infection in children co-infected with *Schistosoma spp* (OR: 0.65; 95%CI: 0.37–1.16). Similar lower odds of *P. falciparum*

**Data Availability Statement:** All relevant data are within the manuscript and its Supporting Information files.

**Funding:** This work was implemented as part of a career development fellowship awarded to MOA, which is funded under UK Research and Innovation Future Leaders Fellowship scheme (MR/S03286X/1). The funders had no role in study design, data collection and analysis, decision to publish, or preparation of the manuscript.

**Competing interests:** The authors have declared that no competing interests exist.

infection were observed from the summary estimates of 24 studies in children co-infected with soil transmitted helminths (STH) (OR: 0.42; 95%CI: 0.28–0.64).

When adjusted for age, gender, socio-economic status, nutritional status and geographic location of the children, the risk of *P. falciparum* infection in children co-infected with STH was higher compared with children who did not have STH infection (OR = 1.3; 95% CI 1.03–1.65).

A subset of 16 studies showed that the odds of anaemia were higher in children co-infected with *Plasmodium* and STH than in children with *Plasmodium* infection alone (OR = 1.20; 95% CI: 0.59–2.45), and were almost equal in children co-infected with *Plasmodium-Schistosoma spp* or *Plasmodium* infection alone (OR = 0.97, 95% CI: 0.30–3.14).

## Conclusions/Significance

The current review suggests that prevalence of malaria-helminth co-infection is high in children living in endemic countries. The nature of the interactions between malaria and helminth infection and the impact of the co-infection on anaemia remain inconclusive and may be modulated by the immune responses of the affected children.

### Author summary

Updated evidence is needed to guide the planning and implementation of appropriate interventions for control of mixed infections involving malaria and worms affecting children living in endemic countries. We performed a systematic review and meta-analysis to update current knowledge on the magnitude of the burden of dual infections with malaria and worms in children in the developing world. We searched all published articles available in Medline, Embase, Global Health and Web of Science from the database inception until 16 March 2020, without any language restriction. We found 55 eligible studies, and 28 of these studies were included in the meta-analysis. A summary of the evidence synthesis showed that the burden of dual infections involving malaria and worm parasites is high in children and varies significantly across endemic countries. There was a lower risk of *P. falciparum* infection in children infected with soil transmitted helminths (STH) or *S. haematobium or S.mansoni*. Conversely, the odds of anaemia were higher in children who had dual infections with P*lasmodium* and STH parasites than in children with a *Plasmodium* infection alone while the odds of anaemia were almost equal in children who were co-infected with *Plasmodium-Schistosoma* compared to those with a *Plasmodium* infection alone. These findings underscore the need to further understand the epidemiology of malaria-helminth co-infections in order to support implementation of appropriate interventions for control and, ultimately, elimination of the dual infections in children living in endemic countries, especially low and middle-income countries (LMIC).

## Introduction

Multi-parasitism, also described as polyparasitism, is the concomitant occurrence of two or more parasite species in a single human host [1]. The parasite species broadly categorised into two groups are macroparasites and microparasites. While macroparasites comprise parasitic

helminths (nematodes and trematodes); microparasites affecting humans are mainly protozoa. Descriptive studies have shown that parasitic helminths (macroparasites) such as soil transmitted helminths (STH) and *Schistosomes*, may co-exist with *Plasmodium* protozoa (microparasites) in children living in resource-poor settings of the world [2–4].

A recent WHO report showed that an estimated 228 million cases of malaria occurred globally, with sub-Saharan Africa (SSA) and Southeast Asia accounting for about 97% of the burden [5]. *Plasmodium falciparum* remains the most prevalent malaria parasite in the WHO African Region, causing 99.7% of the estimated malaria cases in 2018, and 50% in the WHO Southeast Asia Region. Globally, children aged less than five years are the most vulnerable group affected by malaria. In 2018, they accounted for 67% (272 000) of all malaria deaths worldwide [5].

Adding to this very high burden of malaria is the frequent co-existence of parasitic helminths among children living in low and middle-income countries (LMIC) [2]. It is estimated that 1.5 billion individuals are infected with helminths [6], with more than 800 million children in LMIC affected by STH primarily hookworm (*Ancylostoma duodenale* and *Necator americanus*), roundworm (*Ascaris lumbricoides*), and whipworm (*Trichuris trichiura*) [7]. Other important helminths that may co-exist with malaria in children in LMIC include *Schistosoma haematobium* and *S. mansoni* [8].

Schistosomiasis and STH account for a global burden of over 5.2 and 3.3 million disability-adjusted life years, respectively [6] and are associated with anaemia [9, 10], malnutrition [3, 11], and impaired physical and cognitive development [12–14] among preschool-aged children and school-aged children.

An interplay of environmental and host factors has been implicated in favouring mixed infections of parasitic helminths (STH and *Schistosoma* species) with malaria species [15, 16]. Statistical and spatial models also support the geographic overlap and co-endemicity of falciparum malaria and hookworm infections in SSA, suggesting that about 25% of school-aged children are at risk of these two groups of infection [2]. Similar spatial distribution has been documented for the association between malaria and schistosomiasis [11].

This overlap in the endemicity of intestinal helminths and malaria parasites is recognised to be responsible for a high prevalence of malaria-helminth co-infections, with synergistic and antagonistic interactions between helminth and malaria parasites [16–18]. There is some evidence that infections with *Schistosomes* and STH also exert deleterious effects on the course and outcome of clinical malaria [2, 17]. Prominent among the clinical outcomes of malaria-helminth co-infections among children living in LMIC is anaemia [9, 10]. Consequently, in endemic countries where malaria and helminths co-exist, there is potential for these infections to act together to worsen anaemia, a situation previously described as a 'perfect storm of anaemia' [9].

Despite the obvious effects of malaria-helminth co-infections, the nature of the interactions between the two parasites is not clear; existing studies report conflicting findings on the association between malaria and helminths. This may be due to the complexity of multiple pathways involved in the interactions between malaria and helminth parasites during the co-infection [16]. While some studies reported protective effect of hookworms and *S. haematobium* infection against Plasmodium infection [4, 19], other reported increased *Plasmodium* infection in children infected with *S. mansoni* [17].

In 2016, two systematic reviews [7, 8] were published to address the limitations identified in the findings of previous narrative reviews on malaria-helminth co-infections [18, 20]. Consistent with previous reviews, the 2016 review reported an over-estimation of the evidence of the relationship between malaria and helminth parasites making it difficult to conclusively establish the burden of malaria-helminth dual infections and the contribution of helminths to

the interactions in the co-infection [7, 8]. This limitation may be overcome by employing interactive geospatial maps [21–23] to generate real-time epidemiological profile of malaria-helminth co-infections among children in LMIC. This approach has the potential to improve the evidence generated from traditional reviews, which are usually undermined by the methodological limitations of the primary studies included in the reviews.

The objective of this study was to systematically review available data on the burden of malaria-helminth co-infections and the nature of interactions between *Plasmodium* malaria and helminth infections (STH and *Schistosoma spp*) among children living in endemic countries, with a view to establishing how control of malaria and helminth infections might be better co-ordinated in the future.

## Methods

### Protocol development and registration

We developed and prospectively registered the study protocol on PROSPERO (https://www.crd.york.ac.uk/PROSPERO/) [CRD42020171095]. The systematic review was conducted according to PRISMA guidelines [24].

### Eligibility criteria

The PECO framework [25] was used to aid the selection of published articles relevant to the search terms which were eligible for the review, where 'P' stands for the population (children aged 1–16 years living in endemic areas); 'E' for exposure (malaria and helminth co-infection); 'C 'for comparator which is not applicable in the context of this review; and 'O' stands for outcome of interest (anaemia). Epidemiological studies, except case series and case reports, reporting co-infection of *P. falciparum* and/or *P. vivax* with STH (*Ascaris lumbricoides*, hookworm and *T. Trichuria*) and/or schistosomes (*S. haematobium and S.mansoni*), in children aged 1–16 years living in endemic countries, were included. Unpublished research theses, conference abstracts, grey literature, and studies conducted outside endemic countries were excluded after screening the titles and abstracts were excluded. Also excluded were studies which enrolled only adults or pregnant women, because pathophysiology and immunological effects of malaria and helminth infections differ significantly in children from adults and pregnant women [26]. Further studies were removed after review of the full papers, if they did not meet the eligibility criteria highlighted above.

### Definitions

For the purpose of this review, *Plasmodium* infection was defined as microscopic confirmation of asexual *Plasmodium* species with or without clinical features of uncomplicated or severe malaria [27]. *P. falciparum and P. vivax* density was defined as the number of parasites per microliter of blood. STH and *Schistosoma* infections were defined by detection of eggs using FLOTAC, urine filtration or Kato-Katz methods [27]. Kato-Katz thick smears were produced from the stool samples collected from the children and these samples were analysed using light microscopy to determine the egg counts for *S. mansoni* and STH. The numbers of eggs per slide were used to obtain a measure of the number of helminth eggs per gram of faeces (EPG). Anaemia was defined as a haemoglobin concentration below the WHO cut-off values of 11.0 g/dl for children aged 6–59 months; 11.5 g/dl for those 5–11 years; 12.0 g/dl for those 12–14 years [28].

## Search strategy

The first author (MOA) searched Medline, Embase, Global Health and Web of Science from each database inception until 16 March 2020, using compound search strategy combining related truncated and non-truncated terms or synonyms tailored to each of the databases. Additional titles and abstracts were obtained through hand search of a list of references of potentially relevant papers. The search was limited to studies conducted in humans, but no language restrictions were applied. The search terms were 'malaria' OR '*Plasmodium*' OR '*Plasmodium falciparum*' OR '*Plasmodium vivax*' AND 'helminth' OR 'Soil transmitted helminth' OR 'geohelminth' OR '*Ascaris lumbricoides*' OR 'Ascariasis' AND 'hookworm' AND '*Trichuris trichuria*' AND 'bilharziasis' OR 'Schistosoma' OR '*Schistosoma mansoni*' OR '*Schistosoma haematobium*' AND 'children' OR 'paediatric' AND 'developing world' OR 'low and mid-income countries'. Details of the complete search strategy are provided in the protocol (S1 File).

After removing duplicates from the search outputs, screening of 133 abstracts was conducted independently for eligibility for full text review by two reviewers (BMA and EDD), using a systematic review app, Rayyan, (https://rayyan.qcri.org/) [29]. Disagreements between the two reviewers on the inclusion of a study for full text review were resolved following discussions with a third independent reviewer.

## Data collection

Full text papers for 55 eligible abstracts/titles were retrieved and relevant data were extracted. The following data were extracted: first author and year of publication, country/setting, study title and objectives; confirmation of eligibility for review; methodology—study design, sample size, study duration, data collection with study time points; population of interest; prevalence/incidence and density of *Plasmodium* and helminth co-infection, prevalence of anaemia; authors key discussions, comments, limitations and reviewers' comments, where provided.

## Quality and bias assessment

To ascertain the internal and external validity of the included studies, the risk of bias and quality of each study were assessed using a combination of Newcastle Ottawa Scale (NOS) [30] and Effective Public Health Practice Project (EPHHP) tools [31], which are recommended for systematic reviews by The Cochrane Public Health Review Group [32]. Eligible studies were assessed from three angles: selection of the study groups, comparability of the study groups, and the ascertainment of the outcome of interest. Based on these criteria, the overall quality of each study was scored as weak, moderate, or strong. In addition, each study was assessed for selection bias, study design, confounding, blinding, data collection methods, and withdrawals and drop-outs.

## Data analysis

Given a very high level of heterogeneity between included studies, meta-analysis was possible in only 28 studies. Data synthesis was performed to estimate pooled effect estimates across studies, allowing for between-study heterogeneity. We adopted Odds Ratio (OR) as a measure of association. We extracted from the papers data about adjusted odds ratio (aOR) of asymptomatic/uncomplicated or severe *Plasmodium* infection and related anaemia along with 95% confidence interval (CI) for children infected with STH or *S. haematobium* or *S. mansoni*. We extracted raw data for only malaria positive, only STH-positive, only *S. haematobium* or *S. mansoni*-positive, malaria-*S. haematobium* or *S. mansoni*-positive, malaria-and STH-positive,

and intestinal helminth- and malaria-negative and used these data to estimate the pooled prevalence of malaria in the group of children who were malaria positive and STH/*Schistosoma* positive compared to the prevalence in those who were malaria positive and STH/*Schistosoma* negative. We also used the data obtained to calculate the crude odds of asymptomatic/uncomplicated or severe *Plasmodium* infection and related anaemia in children infected with *Schistosomes* or STH compared to those infected with *Plasmodium* parasites only, along with their 95% CI. The log OR and the standard error (SE) of the log OR were estimated using generic inverse variance weighting method [33] and the summary estimate (summary-odds ratio) was estimated. We performed Chi-square heterogeneity test using Cochrane's Q. Moreover, $I^2$ statistic was used to assess the severity of inconsistency (or heterogeneity) across studies. We also examined the studies for publication bias using funnel plots and Egger's test for correlation between the effect estimates and their variances. We used random effects model to estimate the summary Mantel-Haenszel OR of asymptomatic/uncomplicated or severe *Plasmodium* infection in children infected with STH or *Schistosomes* compared to those with *Plasmodium* infection only. We conducted meta-regression to adjust for confounding variables such as age, gender, socio-economic status and nutrition status of the children, as well as study design and geographic location of the study participants. A sensitivity analysis was conducted to assess the robustness of the pooled summary effects by excluding each of the studies from the pooled estimate. Sub-group analyses by helminth types, geographic region and study design were done to compare pooled estimates of the prevalence of the co-infection. All statistical analyses were performed using R software version 4.0.2, R Project for Statistical Computing (https://www.r-project.org/).

Narrative reviews were conducted for 27 studies which did not contain sufficient data that allowed meta-analysis. We adopted a framework consisting of three elements: (i) developing a preliminary synthesis of findings of included studies; (ii) exploring relationships within and between studies; and (iii) assessing the robustness of the synthesis. Studies were grouped together if they compared similar types of outcomes of interest. Content analysis of emerging themes from each group was performed and the summary is presented.

## Results

### Search results and characteristics of included studies

As illustrated in Fig 1, 3,057 citations were generated from database searches on Embase (n = 814), Global Health (n = 1,430), Medline (n = 577), Web of Sciences (n = 220) and manual search from reference list of potentially eligible titles (n = 16). A total of 2,924 duplicates were removed in two stages. One hundred and thirty-three abstracts/titles were screened for eligibility, following which a further 78 citations were excluded due to reasons such as wrong outcomes/wrong populations (n = 67), duplicates (n = 3), and the full text not available (n = 8). We contacted the authors of these abstracts to ascertain whether the full text papers were available. One author confirmed that a manuscript for the abstract was under development, but no feedback was received from the other authors. The full text papers of the remaining 55 citations were reviewed, 27 of which contained inadequate data for meta-analysis. Narrative reviews were performed on these 27 studies [19, 34–59]; data from the remaining 28 studies [4,14,17,60–84] were included in the meta-analysis.

The characteristics of the 55 studies included in the systematic review are summarised in Table 1. A total of 37,559 children were enrolled in the 55 studies conducted across SSA, Southeast Asia and South America. Forty-three of the studies (78.2%) were cross-sectional, five (9.1%) were longitudinal studies, four (7.3%) were randomised controlled trials (RCT), and three (5.4%) were case-control studies. Thirty-seven studies reported *P. falciparum and*

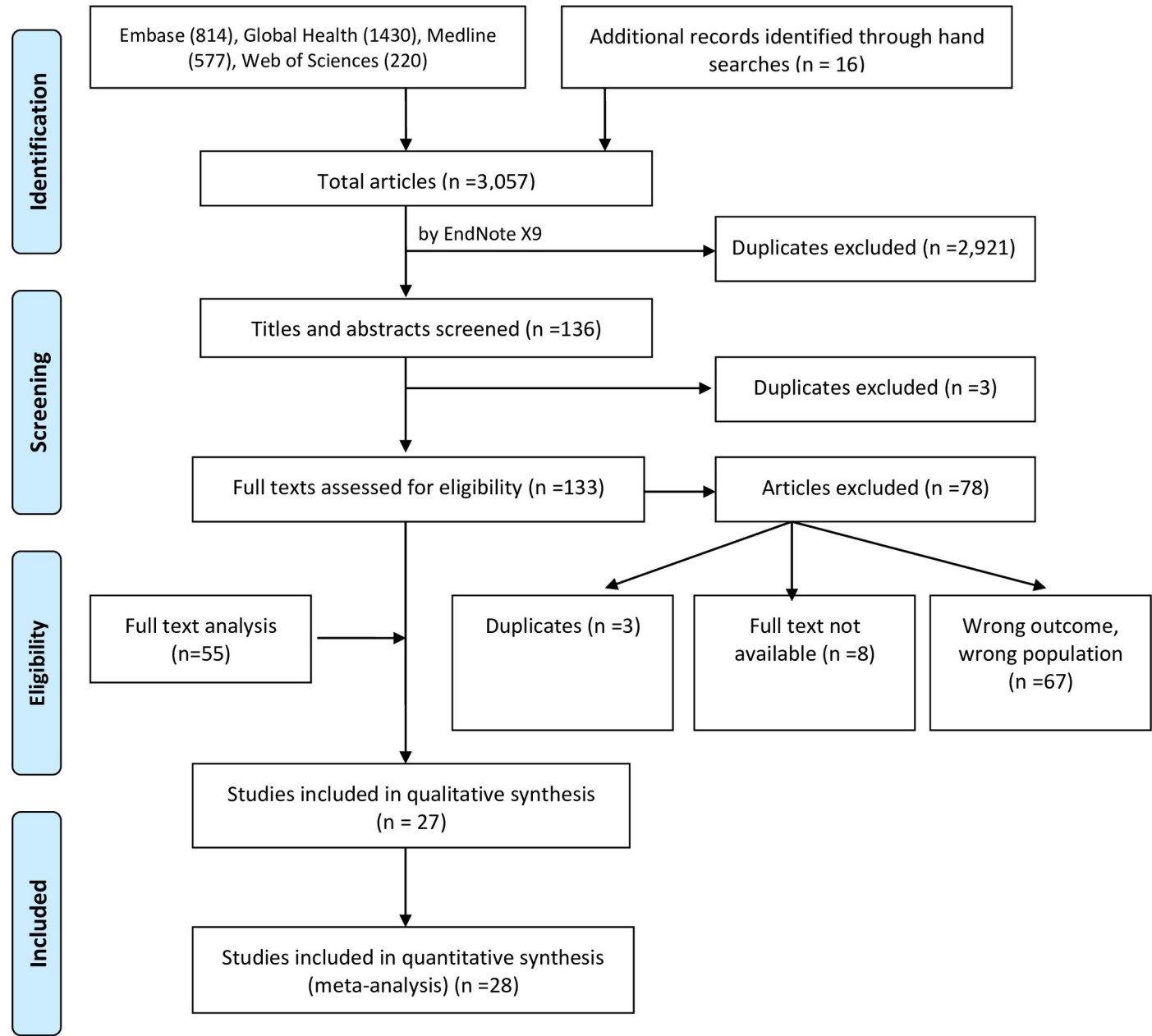

**Fig 1. PRISMA flow diagram showing the process undertaken for inclusion and exclusion of studies in the systemic review.**

STH co-infection; 23 studied *P. falciparum and S. haematobium* co-infection; eight studied *P. falciparum* and *S. mansoni* co-infection and only two reported *P. vivax* and STH co-infection.

Twenty-seven studies compared the odds of asymptomatic/uncomplicated or severe *P. falciparum* infection and 16 studies compared *P. falciparum* density among children infected and uninfected with helminths. Fourteen studies compared the mean haemoglobin concentration or prevalence of anaemia among study participants who had *Plasmodium* and helminth infections and those infected with *Plasmodium* infection only. Most of the studies reported asymptomatic and/or uncomplicated *Plasmodium* infection except three studies [43, 47, 54], which

**Table 1. Characteristics of included studies.**

| SN | Author | Country/year of study | Study design | Sample size | Age range | Co-infection reported | Prevalence of co-infection (%) | Magnitude of outcomes in *Plasmodium*-helminth co-infected participants vs those infected with *Plasmodium* only | Overall quality score |
|----|--------|----------------------|--------------|-------------|-----------|----------------------|-------------------------------|--------------------------------------------------------------------------------------------------------------------|----------------------|
| 1 | Achidi et al, 2008 [60] | Cameroon 2004–2005 | Cross-sectional | 263 | 4–12 years | Pf+ STH | 65.9 | 1. Similar Pf prevalence (OR: 1.29; 95% CI 0.60–2.76) 2. Similar Hb levels | Moderate |
| 2 | Adedoja et al, 2015 [61] | Nigeria 2012–2013 | Cross-sectional | 1017 | 4–15 years | Pf+ HK Pf + Sh | 29.2 50.7 | 1.Higher Pf prevalence 2. Similar Pf risk: Children infected with Sh (RR = 1.3, p = 0.04) and HK (RR = 1.4, p = 0.01) have equal chances of being infected with Pf 3. Higher anaemia risk | Strong |
| 3 | Doumbo et al, 2014 [67] | Mali 2011–2012 | Longitudinal cohort | 616 | 3 months—25 years | Pf + Sh | 8.5 | 1. Higher Pf prevalence, OR: 3.23; 95% CI: 1.76, 5.90 2. Similar Pf risk (HR) 3. Heavy Sh negatively associated with Pf density but light Sh not associated with Pf density | Strong |
| 4 | Akanni et al, 2014 [34] | Nigeria NR | Cross-sectional | 292 | 1–15 years | Pf + STH Pf+ Sm | NR | Lower mean Hb and serum ferritin levels (p>0.05) | Weak |
| 5 | Humphries et al, 2013 [36] | Ghana 2010 | Cross-sectional | 812 | 6–11 years | Pf+ HK | 35.1 | Higher Pf density, with OR for HK infection increasing with higher Pf parasite density (OR = 5.50, 95% CI 2.10, 14.41, P< 0.01) | Moderate |
| 6 | Humphries et al, 2011 [35] | Ghana 2007 | Cross-sectional | 132 | 1–15 years | Pf + HK | 51% in 6–10 year olds | 1. Higher Pf prevalence (aOR = 2.84, 95%CI = 1.11, 7.26), with highest prevalence of co-infection (51%) in 6–10 year olds. 2. Similar anaemia risk (aOR = 2.34; 95% CI = 0.56, 9.84, p = 0.25) | Moderate |
| 7 | Ajayi et al, 2015 [84] | Nigeria 2011 | Cross-sectional | 370 | 6 months—14 years | Pf+ STH | 42.9 | 1.Risk of Pf +STH co-infection about two times in children aged 5.5–10 years compared to under-fives (OR = 2.3, 95% CI = 0.19, P = 0.95). 2. Similar anaemia risk (P = 0.93) | Weak |
| 8 | Briand et al, 2005 [37] | Senegal 2001–2002 | Cross-sectional | 523 | 3–15 years | Pf + Sh | NR | Children lightly infected with Sh had lower Pf densities than non-infected children (β: -0.34, 95% CI: −0.85, −0.10) | Moderate |
| 9 | Burdam et al, 2016 [64] | Indonesia 2013 | Cross-sectional | 533 | 12–59 months | Pf+ STH Pv + STH | 7.1 15.3 | 1. Higher Plasmodium risk = Pv: OR 3.75 (95%CI, 1.53–9.2), p = 0.004; Pf: OR 2.0 (95%CI, 0.4–10.1), p = 0.402 2. Higher anaemia risk for Pf or Pv + STH (OR 4.0 [95%CI, 1.4–11.3], p = 0.008) and severe anaemia (OR 8.6 [95%CI, 1.3–55.8], p = 0.024). | Moderate |
| 10 | Brutus et al, 2006 [38] | Madagascar 1996–1997 | RCT | 350 | 4–16 years | Pf+ Al | NR | 1. Levamisole treatment effect on Al + Pf = 0.58, 95% CI: 0.10–0.95, p = 0.018 2. Negative interaction between Al and Pf parasite density in children aged ≥5 years | Moderate |

*(Continued)*

**Table 1.** (*Continued*)

| SN | Author | Country/year of study | Study design | Sample size | Age range | Co-infection reported | Prevalence of co-infection (%) | Magnitude of outcomes in *Plasmodium*-helminth co-infected participants vs those infected with *Plasmodium* only | Overall quality score |
|----|--------|----------------------|--------------|-------------|-----------|----------------------|-------------------------------|---------------------------------------------------------------------------------------------------------------------|----------------------|
| 11 | Brutus et al, 2007 [39] | Madagascar 1996–1997 | RCT | 312 | 4–16 years | Pf+ Al | NR | 1. Levamisole treatment effect on Al + Pf = 0.58, 95% CI: 0.20–0.95, p = 0.003<br>2. Negative interaction between Al infection and Pf parasite density in 5–14 year old | Moderate |
| 12 | Bwanika et al, 2018 [65] | Uganda NR | Case-control | 240 | 7.6 ± 3.0 years | Pf + STH<br>Pf+ Al<br>Pf+ HK | 13.3<br>78.1<br>21.2 | Higher levels of IL-10 (304 pg/ml) expressed compared to Pf only infected children (11.57 pg/ml), P< 0.05 | Moderate |
| 13 | Carmona_Fonseca et al, 2006 [66] | Colombia 2004–2005 | Cross-sectional | 93 | 4–10 years | Pf + Al + HK+ Tt | 74–97 | 1. Similar Pf prevalence. 2. Frequency of associations not increased with increasing parasite loads | Weak |
| 14 | Dejon_Agobe et al_2018 [4] | Gabon 2012–2014 | Longitudinal survey | 739 | NR | Pf+ Al<br>Pf+ HK<br>Pf+ Tt<br>Pf+ Sh | 23.6<br>25.6<br>29.5<br>9 | 1.Higher Pf incidence, with a significant delay of time-to first-malaria event only in children aged 6-10-years-old infected with Sh<br>2. Higher Pf risk | Strong |
| 15 | Degarege et al, 2014 [81] | Ethiopia 2010–2011 | Cross-sectional | 702 | NR | Pf + STH | 19.4 | Similar Pf prevalence and risk of undernutrition | Moderate |
| 16 | Deribew et al, 2013 [40] | Ethiopia 2013 | cross-sectional | 387 | 6–23 months | Pf+ Sh | 2.84 | 1. Higher Pf prevalence, OR: 2.8; 95% CI: 1.21–6.5<br>2. Higher anaemia prevalence, OR: 10.12; 95% CI: 1.47–69.94<br>3. Similar mean Hb difference | Weak |
| 17 | Doumbo et al, 2018 [68] | Mali 2011 | Cross-sectional | 688 | 3 months-25years | Pf+ Sh | 2.18 | Higher Pf prevalence | Moderate |
| 18 | Elfaki et al, 2015 [69] | Sudan 2008–2009 | Cross-sectional | 250 | 6–16 years | Pf+ Sh<br>Pf+ Sm | 6.4<br>4 | Higher Pf prevalence | Weak |
| 19 | Green et al, 2011 [41] | Uganda 2009 | Cross-sectional | 573 | 0.4–6 years | Pf+ Sm<br>Pf + STH | 28.7<br>1.8 | Similar prevalence of anaemia and Hb levels | Moderate |
|    |        |                      |              | 453 | 2–6 years | Pf+ Sm<br>Pf + STH | 20.1<br>6.3 |  |  |
| 20 | Kepha et al, 2014 [42] | Kenya 2013 | Cross-sectional | 5471 | 5–18 years | Pf+ Al<br>Pf+ HK | 7.8<br>9 | Higher Pf risk when adjusted for age and gender | Moderate |
| 21 | Kinugh'hi et al, 2014 [71] | Tanzania 2006 | Cross-sectional | 1546 | 3–13 years | Pf+ HK<br>Pf+ Sh | 15.8<br>10.2 | Similar Pf prevalence (OR: 1.19; 95% CI: 0.86–1.63) | Moderate |
| 22 | Kwenti et al, 2016 [43] | Cameroon 2014–2015 | Cross-sectional | 405 | 1 week-120 months | Pf + STH | 11.6 | Similar Pf prevalence | Moderate |
| 23 | Salim et al, 2015 [77] | Tanzania 2011–2012 | Cross-sectional | 992 | 6 months -9 years | Pf+ HK<br>Pf+ Tt<br>Pf + Sh | 1.4<br>0.1<br>0.2 | Higher Pf prevalence and risk (OR = 1.7, 95% CI = 1.1–2.5 | Moderate |
| 24 | Matangila et al, 2014 [72] | DR Congo 2012 | Cross-sectional | 989 | 4–13 years | Pf+ STH<br>Pf+ Sm | 6.4<br>1.5 | 1.Similar Pf prevalence<br>2. Lower anaemia prevalence | Moderate |
| 25 | Mazigo et al, 2010 [73] | Tanzania 2009 | Cross-sectional | 400 | 8–16 years | Pf+ HK<br>Pf + Sm | 2.3<br>5.5 | 1. Similar Pf prevalence (aOR: 1.35; 95% CI: 0.49–3.72)<br>2. Higher Pf prevalence OR: 2.1; 95% CI: 1.03, 4.26 | Moderate |
| 26 | Mboera et al, 2011 [83] | Tanzania 2005 | Cross-sectional | 578 | 7.96 ± 1.4 years | Pf + HK<br>Pf + Sh | 5.3<br>10.9 | Higher Pf density (P < 0.001) | Moderate |

(*Continued*)

**Table 1.** (*Continued*)

| SN | Author | Country/year of study | Study design | Sample size | Age range | Co-infection reported | Prevalence of co-infection (%) | Magnitude of outcomes in *Plasmodium*-helminth co-infected participants vs those infected with *Plasmodium* only | Overall quality score |
|---|---|---|---|---|---|---|---|---|---|
| 27 | Morenikeji et al, 2016 [74] | Nigeria NR | Cross-sectional | 173 | 6–18 years | Pf + Sh | 62.3 | Similar Pf risk OR: 0.8, 95% CI: 0.3–1.7, P = 0.507 | Moderate |
| 28 | Nankabirwa et al, 2013 [14] | Uganda 2011 | Cross-sectional | 740 | 6–14 years | Pf +Al Pf + HK Pf + Tt | 28.9 40 66.7 | Similar Pf prevalence | Moderate |
| 29 | Njunda et al, 2015 [44] | Cameroon 2012 | Cross-sectional | 411 | 0–10 years | Pf + STH | 11.9 | 1.Similar Pf prevalence 2.Similar anaemia prevalence and mean Hb difference | Moderate |
| 30 | Nkuo-Akenji et al, 2006 [75] | Cameroon 2004 | Cross-sectional | 425 | 0–14 years | Pf+ Al Pf + HK Pf + Tt | 7.5 0.2 7.8 | 1. Similar Pf prevalence (OR: 1.00; 95% CI: 0.65–1.53) 2. Higher Pf density (P < 0.001) 3. Similar anaemia prevalence (OR: 1.19; 95% CI: 0.642.21) | Moderate |
| 31 | Nyarko et al, 2018 [45] | Ghana NR | Cross-sectional | 404 | 9–14 years | Pf + Sh | 12.9 | Similar Pf prevalence | Moderate |
| 32 | Obi et al, 1996 [76] | Nigeria NR | Cross-sectional | 268 | 5–15 years | Pf + Sh | 21.3 | NR | Weak |
| 33 | Ojurongbe et al, 2011 [82] | Nigeria 2009 | Cross-sectional | 117 | 4–15 years | Pf + Al Pf + HK Pf + Tt | 13.3 6.7 3.3 | 1.Higher Pf prevalence 2. Similar anaemia prevalence | Moderate |
| 34 | Tchinda et al, 2012 [78] | Cameroon NR | Cross-sectional | 503 | 3–16 years | Pf + Al | 49.6 | 1.Higher Pf prevalence: OR = 1.61, 95% CI = 1.05–2.44, P = 0.028 2.Higher anaemia risk: OR = 1.64, 95% CI: 0.98–2.75; p = 0.059 | Moderate |
| 35 | Yapi et al, 2014 [79] | Senegal 2011–2012 | Cross-sectional | 5104 | 5–16 years | Pf + STH Pf + Sh | 13.5 5.6 | Higher Pf risk: IRR = 0.7, 95% CI = 0.6–0.8, P = 0.001 | Strong |
| 36 | Zeukeng et al, 2014 [80] | Cameroon 2011 | Cross-sectional | 152 | 1–14 years | Pf + Al Pf+ HK Pf + Tt | 21.1 4.61 8.7 | 1.Higher Pf prevalence 2.Higher anaemia prevalence | Moderate |
| 37 | Njua-Yafi et al, 2016 [46] | Cameroon 2011–2012 | Cross-sectional | 357 | 6 months -10 years | Pf + STH | 2.9 | 1.Higher Pf prevalence: OR = 4.45, 95% CI (1.66–11.94), p = 0.003 2.Higher mean Hb levels (p<0.006) | Moderate |
| 38 | Le Hesran et al, 2004 [47] | Senegal 2001–2002 | Case-control | 105 | Mean age = 6.6 ±3 years | Pf + Sh | 10.3 | 1.Higher Pf risk: OR = 9.95, 95% CI = 3.03–32.69 | Weak |
| 39 | Lemaitre et al, 2014 [48] | Senegal 2001–2003 | Cross-sectional | 523 | 5–13 years | Pf + Sh | NR | Higher Pf density: OR = -0.28, 95% CI: −0.52, −0.039, P = 0.10 | Weak |
| 40 | Nacher et al, 2012 [49] | Thailand 1998–1999 | Cross-sectional | 731 | Median age = 17 years | Pf + STH | 12.7 | Higher Pf risk | Moderate |
| 41 | Stefani et al, 2017 [50] | French Guiana 2001–2009 | Cross-sectional | 91 | 0–7 years | Pv+ HK Pv + Al | 35.5 46.5 | Higher Pv relapse: OR = 0.36, 95% CI = 0.94–1.4, p = 0.09 | Moderate |
| 42 | Lyke et al, 2005 [51] | Mali 2002–2003 | Case-control | 654 | 4–14 years | Pf + Sh | 44.5 | 1. Lower Pf incidence in 4–8 y.o 2. Lower Pf density in children ages 4–8 y.o 3. Similar Hb level | Moderate |

(*Continued*)

**Table 1.** (*Continued*)

| SN | Author | Country/year of study | Study design | Sample size | Age range | Co-infection reported | Prevalence of co-infection (%) | Magnitude of outcomes in *Plasmodium*-helminth co-infected participants vs those infected with *Plasmodium* only | Overall quality score |
|---|---|---|---|---|---|---|---|---|---|
| 43 | Sangweme et al, 2010 [17] | Zimbabwe 2004–2005 | Longitudinal cohort | 605 | 6–17 years | Pf+ Sh | 30.7 | 1. Similar Pf prevalence<br>2. Lower Pf density<br>3. Similar mean Hb<br>4. Similar anaemia prevalence | Moderate |
| 44 | Adio et al, 2004 [62] | Nigeria 2002 | Cross-sectional | 243 | 6 months -15 years | Pf + Al<br>Pf + HK<br>Pf +Tt | 26.3<br>16.5<br>20.2 | 1.Higher Pf density<br>2.Higher anaemia prevalence | Weak |
| 45 | Adedoja et al, 2015 [52] | Nigeria 2003 | Cross-sectional | 154 | 1–15 years | Pf + Sh | 54.4 | Higher Pf prevalence | Weak |
| 46 | Righetti et al, 2011 [53] | Ivory Coast 2010 | Cross sectional | 147 | 5–14 years | Pf + HK<br>Pf + Sh | 27.9<br>8.8 | 1. Higher Pf prevalence (in 6–8 year-old) (aOR: 7.47–95% CI: 1.84–30.32)<br>2. Lower anaemia prevalence (OR: 0.23; 95% CI: 0.06–0.83) | Moderate |
| 47 | Abanyie et al, 2013 [54] | Nigeria 2006–2007 | RCT | 690 | 6–59 months | Pf + Al | 42.9 | 1. Similar Pf prevalence (OR: 1.30; 95% CI: 0.91–1.86)<br>2. Lower anaemia prevalence (OR: 0.50; 95% CI: 0.28–0.87); severity of malarial anaemia not altered by Pf-Al co-infection<br>3. Similar Pf density (p = 0.965) | Moderate |
| 48 | Roussihon et al, 2010 [55] | Senegal 2000–2005 | Longitudinal study | 203 | 1–14 years | Pf+ STH | 0.655/child/year; ratio of malaria attack incidence density = 1: 1.723 | 1.Higher Pf incidence (aOR: 2.69; 95% CI: 1.34–5.39)<br>2.Incidence of malaria attacks is increased in helminth-infected children | Moderate |
| 49 | Kirwan et al, 2010 [56] | Nigeria 2006–2007 | RCT | 320 | 6–59 months | Pf + Al | NR | 1. Similar Pf prevalence (OR: 1.16; 95% CI: 0.73–1.85)<br>2. Similar rate of increase in Pf density<br>3. Similar rate of increase in Hb levels<br>4. Repeated 4-monthly administration of anti-helminth lowers Pf prevalence, coincided with a reduction in both Al prevalence and intensity | Strong |
| 50 | Ateba-Ngoa et al, 2015 [57] | Gabon 2011 | Cross-sectional | 125 | 6–16 years | Pf+ Sh | 26 | 1. Similar magnitude between the two groups<br>2. Higher mean Hb difference: OR = 0.7; 95% CI: 0.21, 1.19<br>3. Increased immune response due to Pf | Moderate |
| 51 | Florey et al, 2012 [58] | Kenya 2006 | Cross-sectional | 223 | 8–17 years | Pf + Sh | 36.3<br>15.4 | 1. Higher Pf prevalence, adjusted OR: 1.79; 95% CI: 1.32, 2.44<br>2. Children with heavy Pf more likely to have concomitant heavy Sh | Moderate |
| 52 | Courtin et al, 2011 [59] | Senegal 2003 | Longitudinal survey | 203 | 6–16 years | Pf+ Sh | NR | 1. Similar Pf prevalence, OR: 1.62; 95% CI: 0.94, 2.80<br>2. Similar Pf density<br>3. Additive effect of S. haem and Pf on the cytokine levels (elevated IL-10) | Moderate |

(*Continued*)

**Table 1.** (Continued)

| SN | Author | Country/year of study | Study design | Sample size | Age range | Co-infection reported | Prevalence of co-infection (%) | Magnitude of outcomes in *Plasmodium*-helminth co-infected participants vs those infected with *Plasmodium* only | Overall quality score |
|---|---|---|---|---|---|---|---|---|---|
| 53 | Kabatereine et al, 2011 [70] | Uganda 2009–2010 | Cross-sectional | 3569 | 10–14 years | Pf + Sm Pf + STH | 23.51 13.79 | Higher Pf prevalence, OR: 2.16; 95% CI: 1.89, 2.47 | Moderate |
| 54 | Alemu et al, 2012 [63] | Ethiopia 2011 | Cross-sectional | 108 | 1–14 years | Pf + STH | 30 | 1.Higher Pf prevalence 2.Higher anaemia prevalence: OR = 13.800, 95% CI = 4.871–39.093, P<0.0001 | Moderate |
| 55 | Sokhna et al, 2004 [19] | Senegal, 2008 | Cross-sectional | 512 | 6–15 years | Pf + Sm | 13.3 | Higher Pf incidence in children with heavy S. *mansoni* egg load compared with uninfected (RR: 2.24, 95% CI:1.20, 4.20) | Moderate |

reported severe malaria. Seven studies [19, 37, 53, 55, 57, 59, 65] reported antibody and/or cytokine expression in *P. falciparum* malaria-STH/*Schistosoma* co-infections.

## Prevalence of *Plasmodium*-helminth co-infections

Overall, the pooled analysis showed that the prevalence of *Plasmodium*-STH co-infections in 22,114 children enrolled in 28 studies evaluating both groups of infection, in 13 countries across Central, East, West and Southern Africa, Southeast Asia and South America was 17.7% (95% CI 12.7–23.2%) (Fig 2). There was statistical heterogeneity ($I^2$: 98.7%), suggesting a high level of heterogeneity among the included studies, although Egger's regression test did not reach statistical significance (Egger's test = 1.5039, p-value = 0.1388) (Fig 3). In a subgroup analysis, the pooled estimates showed that South America had the highest prevalence of 76.3% of co-infections, followed by Southern Africa (30.7%), West Africa (25.4%), East Africa (16.3%), Central Africa (13.1%) and South-east Asia (7.1%) (S1 Fig). Disaggregated by helminth types, the pooled prevalence of *Plasmodium-Schistosoma* co-infections in 9,803 children was 19.2% (95% CI: 9.6–31.1%); and the pooled prevalence of *Plasmodium*-STH co-infections in 12311 children was 17% (95%CI: 11.4–23.4%); (95% CI: 9.6–31.1%) (Fig 2). In a subgroup analysis of 12 studies reporting co-infection of *Plasmodium* and individual STH types in 6325 children, the prevalence for *P.falciparum-A. lumbricoides* co-infection was 24% (95% CI: 11.3–39.6%), *P.falciparum*-hookworm 12.8% (95% CI: 4.8–23.9%) and *P. falciparum-T.trichuria* 9.9% (95% CI: 2.6–21.4%) (S2A, S2B and S2C Fig, respectively). For *Plasmodium falciparum-S. haematobium* co-infection, the pooled prevalence was 24.4% (95% CI: 11.9–39.6%) and 6.7% (95% CI: 1.8–14.6%) for *Plasmodium falciparum-S. mansoni* co-infection (S3A and S3B Fig, respectively). The funnel plot generated using the cases of *Plasmodium*-helminth co-infections compared with children infected with *Plasmodium* mono-infection and the standard error estimates of the prevalence was asymmetric, in keeping with the high level of heterogeneity between the included studies, but as indicated above, the Egger's regression test was not statistically significant (Egger's test = 1.5039, p-value = 0.1388) (Fig 3).

## Interactions of *Plasmodium* and helminths during co-infection

Given a very high level of heterogeneity between the studies, random-effects models were used to calculate the summary odds ratio in this meta-analysis. The overall estimates on 24 studies showed lower odds of asymptomatic/uncomplicated or severe *P. falciparum* infection in children co-infected with STH when compared with those with *Plasmodium* only (summary OR:

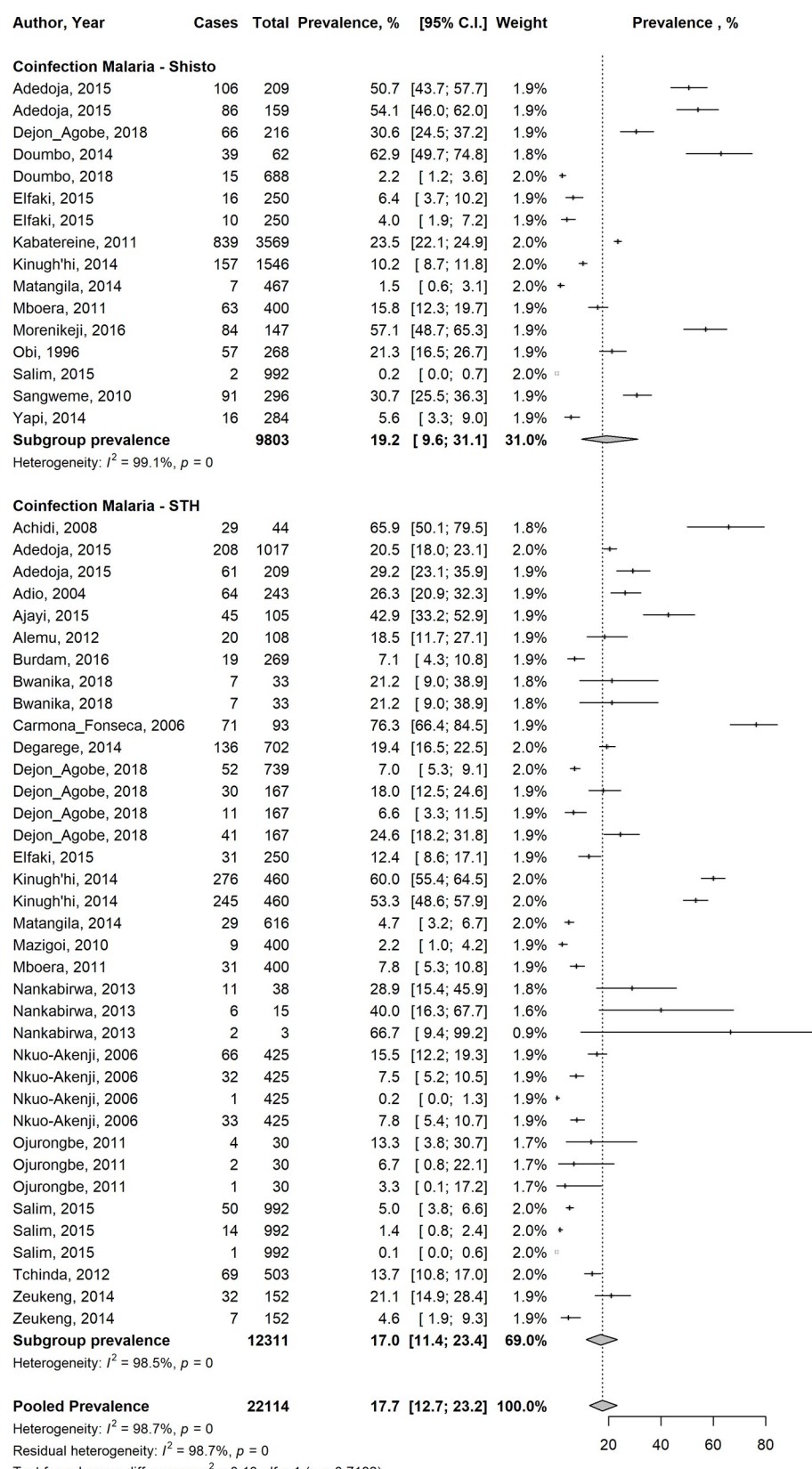

| Author, Year | Cases | Total | Prevalence, % | [95% C.I.] | Weight |
|---|---|---|---|---|---|
| **Coinfection Malaria - Shisto** | | | | | |
| Adedoja, 2015 | 106 | 209 | 50.7 | [43.7; 57.7] | 1.9% |
| Adedoja, 2015 | 86 | 159 | 54.1 | [46.0; 62.0] | 1.9% |
| Dejon_Agobe, 2018 | 66 | 216 | 30.6 | [24.5; 37.2] | 1.9% |
| Doumbo, 2014 | 39 | 62 | 62.9 | [49.7; 74.8] | 1.8% |
| Doumbo, 2018 | 15 | 688 | 2.2 | [1.2; 3.6] | 2.0% |
| Elfaki, 2015 | 16 | 250 | 6.4 | [3.7; 10.2] | 1.9% |
| Elfaki, 2015 | 10 | 250 | 4.0 | [1.9; 7.2] | 1.9% |
| Kabatereine, 2011 | 839 | 3569 | 23.5 | [22.1; 24.9] | 2.0% |
| Kinugh'hi, 2014 | 157 | 1546 | 10.2 | [8.7; 11.8] | 2.0% |
| Matangila, 2014 | 7 | 467 | 1.5 | [0.6; 3.1] | 2.0% |
| Mboera, 2011 | 63 | 400 | 15.8 | [12.3; 19.7] | 1.9% |
| Morenikeji, 2016 | 84 | 147 | 57.1 | [48.7; 65.3] | 1.9% |
| Obi, 1996 | 57 | 268 | 21.3 | [16.5; 26.7] | 1.9% |
| Salim, 2015 | 2 | 992 | 0.2 | [0.0; 0.7] | 2.0% |
| Sangweme, 2010 | 91 | 296 | 30.7 | [25.5; 36.3] | 1.9% |
| Yapi, 2014 | 16 | 284 | 5.6 | [3.3; 9.0] | 1.9% |
| **Subgroup prevalence** | | **9803** | **19.2** | **[9.6; 31.1]** | **31.0%** |
| Heterogeneity: $I^2$ = 99.1%, $p$ = 0 | | | | | |
| | | | | | |
| **Coinfection Malaria - STH** | | | | | |
| Achidi, 2008 | 29 | 44 | 65.9 | [50.1; 79.5] | 1.8% |
| Adedoja, 2015 | 208 | 1017 | 20.5 | [18.0; 23.1] | 2.0% |
| Adedoja, 2015 | 61 | 209 | 29.2 | [23.1; 35.9] | 1.9% |
| Adio, 2004 | 64 | 243 | 26.3 | [20.9; 32.3] | 1.9% |
| Ajayi, 2015 | 45 | 105 | 42.9 | [33.2; 52.9] | 1.9% |
| Alemu, 2012 | 20 | 108 | 18.5 | [11.7; 27.1] | 1.9% |
| Burdam, 2016 | 19 | 269 | 7.1 | [4.3; 10.8] | 1.9% |
| Bwanika, 2018 | 7 | 33 | 21.2 | [9.0; 38.9] | 1.8% |
| Bwanika, 2018 | 7 | 33 | 21.2 | [9.0; 38.9] | 1.8% |
| Carmona_Fonseca, 2006 | 71 | 93 | 76.3 | [66.4; 84.5] | 1.9% |
| Degarege, 2014 | 136 | 702 | 19.4 | [16.5; 22.5] | 2.0% |
| Dejon_Agobe, 2018 | 52 | 739 | 7.0 | [5.3; 9.1] | 2.0% |
| Dejon_Agobe, 2018 | 30 | 167 | 18.0 | [12.5; 24.6] | 1.9% |
| Dejon_Agobe, 2018 | 11 | 167 | 6.6 | [3.3; 11.5] | 1.9% |
| Dejon_Agobe, 2018 | 41 | 167 | 24.6 | [18.2; 31.8] | 1.9% |
| Elfaki, 2015 | 31 | 250 | 12.4 | [8.6; 17.1] | 1.9% |
| Kinugh'hi, 2014 | 276 | 460 | 60.0 | [55.4; 64.5] | 2.0% |
| Kinugh'hi, 2014 | 245 | 460 | 53.3 | [48.6; 57.9] | 2.0% |
| Matangila, 2014 | 29 | 616 | 4.7 | [3.2; 6.7] | 2.0% |
| Mazigoi, 2010 | 9 | 400 | 2.2 | [1.0; 4.2] | 1.9% |
| Mboera, 2011 | 31 | 400 | 7.8 | [5.3; 10.8] | 1.9% |
| Nankabirwa, 2013 | 11 | 38 | 28.9 | [15.4; 45.9] | 1.8% |
| Nankabirwa, 2013 | 6 | 15 | 40.0 | [16.3; 67.7] | 1.6% |
| Nankabirwa, 2013 | 2 | 3 | 66.7 | [9.4; 99.2] | 0.9% |
| Nkuo-Akenji, 2006 | 66 | 425 | 15.5 | [12.2; 19.3] | 1.9% |
| Nkuo-Akenji, 2006 | 32 | 425 | 7.5 | [5.2; 10.5] | 1.9% |
| Nkuo-Akenji, 2006 | 1 | 425 | 0.2 | [0.0; 1.3] | 1.9% |
| Nkuo-Akenji, 2006 | 33 | 425 | 7.8 | [5.4; 10.7] | 1.9% |
| Ojurongbe, 2011 | 4 | 30 | 13.3 | [3.8; 30.7] | 1.7% |
| Ojurongbe, 2011 | 2 | 30 | 6.7 | [0.8; 22.1] | 1.7% |
| Ojurongbe, 2011 | 1 | 30 | 3.3 | [0.1; 17.2] | 1.7% |
| Salim, 2015 | 50 | 992 | 5.0 | [3.8; 6.6] | 2.0% |
| Salim, 2015 | 14 | 992 | 1.4 | [0.8; 2.4] | 2.0% |
| Salim, 2015 | 1 | 992 | 0.1 | [0.0; 0.6] | 2.0% |
| Tchinda, 2012 | 69 | 503 | 13.7 | [10.8; 17.0] | 2.0% |
| Zeukeng, 2014 | 32 | 152 | 21.1 | [14.9; 28.4] | 1.9% |
| Zeukeng, 2014 | 7 | 152 | 4.6 | [1.9; 9.3] | 1.9% |
| **Subgroup prevalence** | | **12311** | **17.0** | **[11.4; 23.4]** | **69.0%** |
| Heterogeneity: $I^2$ = 98.5%, $p$ = 0 | | | | | |
| | | | | | |
| **Pooled Prevalence** | | **22114** | **17.7** | **[12.7; 23.2]** | **100.0%** |
| Heterogeneity: $I^2$ = 98.7%, $p$ = 0 | | | | | |
| Residual heterogeneity: $I^2$ = 98.7%, $p$ = 0 | | | | | |
| Test for subgroup differences: $\chi_1^2$ = 0.13, df = 1 ($p$ = 0.7182) | | | | | |

**Fig 2. Forest plot showing overall pooled prevalence of *Plasmodium*-helminth co-infections among children living in LMIC in studies which evaluated the prevalence of both groups of infection.**

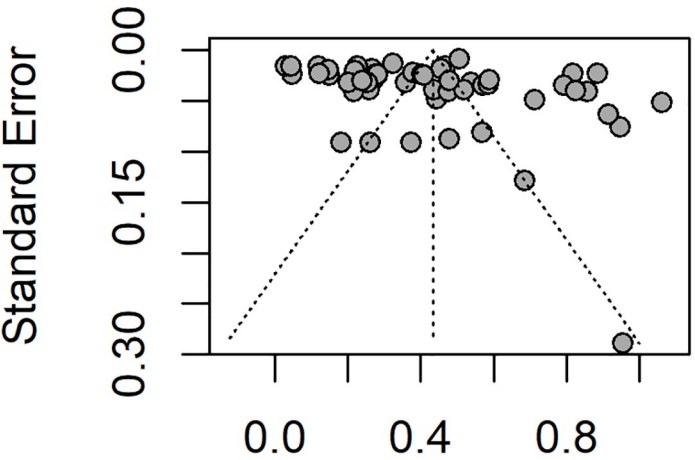

**Fig 3. Funnel plot showing distributions of included studies.**

0.42; 95%CI: 0.28–0.64; $I^2$ = 96.2%) (Fig 4). Similarly, the overall estimates in 20 studies showed lower odds of asymptomatic/uncomplicated *P. falciparum* infection among children co-infected with *Schistosoma spp*. compared with those with *Plasmodium* infection only (summary OR: 0.65; 95%CI: 0.37–1.16; $I^2$ = 96.8%), although the odds-ratio did not reach statistical significance (Fig 5). Compared with children who had *Plasmodium* infection only, sub-group analyses showed similar lower odds in the summary estimates in children who had co-infection with *Plasmodium*-STH sub-types: *A. lumbricoides* - 0R =: 0.39, 95% CI: 0.12–1.23, $I^2$ = 96.1%); hookworm–OR = 0.31, 95%CI 0.12–0.80, $I^2$ = 96.8%); *T.trichuria*–(OR = 0.15, 95% CI 0.03–0.77; $I^2$ = 97.8%) (S2A, S2B and S2C Fig, respectively). For *Plasmodium-S. haematobium* and *Plasmodium-S. mansoni, the* OR was 0.76, 95% CI: 0.41–1.41; $I^2$ = 99.1% and OR:0.39, 95% CI: 0.1–1.58, $I^2$ = 99.0% respectively (S3A and S3B Fig, respectively). When adjusted for age, gender, socio-economic status, nutrition status and geographic locations of the children, the risk of asymptomatic/uncomplicated or severe *P. falciparum* infection in children co-infected with STH was higher compared with children who did not have STH infection (OR = 1.3, 95% CI 1.03–1.65; $I^2$ = 32.1%). (S4 and S5 Figs).

### Risk of Anaemia in *Plasmodium*-helminth co-infections

In 15 studies [35,44,46,52–54,62–64,71,75,78,80,83–84], the odds of anaemia were higher in children who were co-infected with *Plasmodium* species and pooled STH species compared to those who were infected with *P. falciparum* alone (summary OR = 1.20 (95% CI: 0.59–2.45, $I^2$ = 96.3%) (Fig 6). In contrast, the odds of anaemia were almost equal in six studies [17, 40, 61, 71, 72, 83] involving children who were co-infected with *Plasmodium* species and pooled *Schistosoma* species compared to those who were infected with *P. falciparum* alone (summary OR = 0.97, 95% CI: 0.30–3.14; $I^2$ = 97.8%) (Fig 7).

### Narrative review

**Roles of biologic factors on malaria-helminth co-infection.**   Four studies [42,43,52,58] explored the roles of biological factors such as age and gender in the epidemiology of

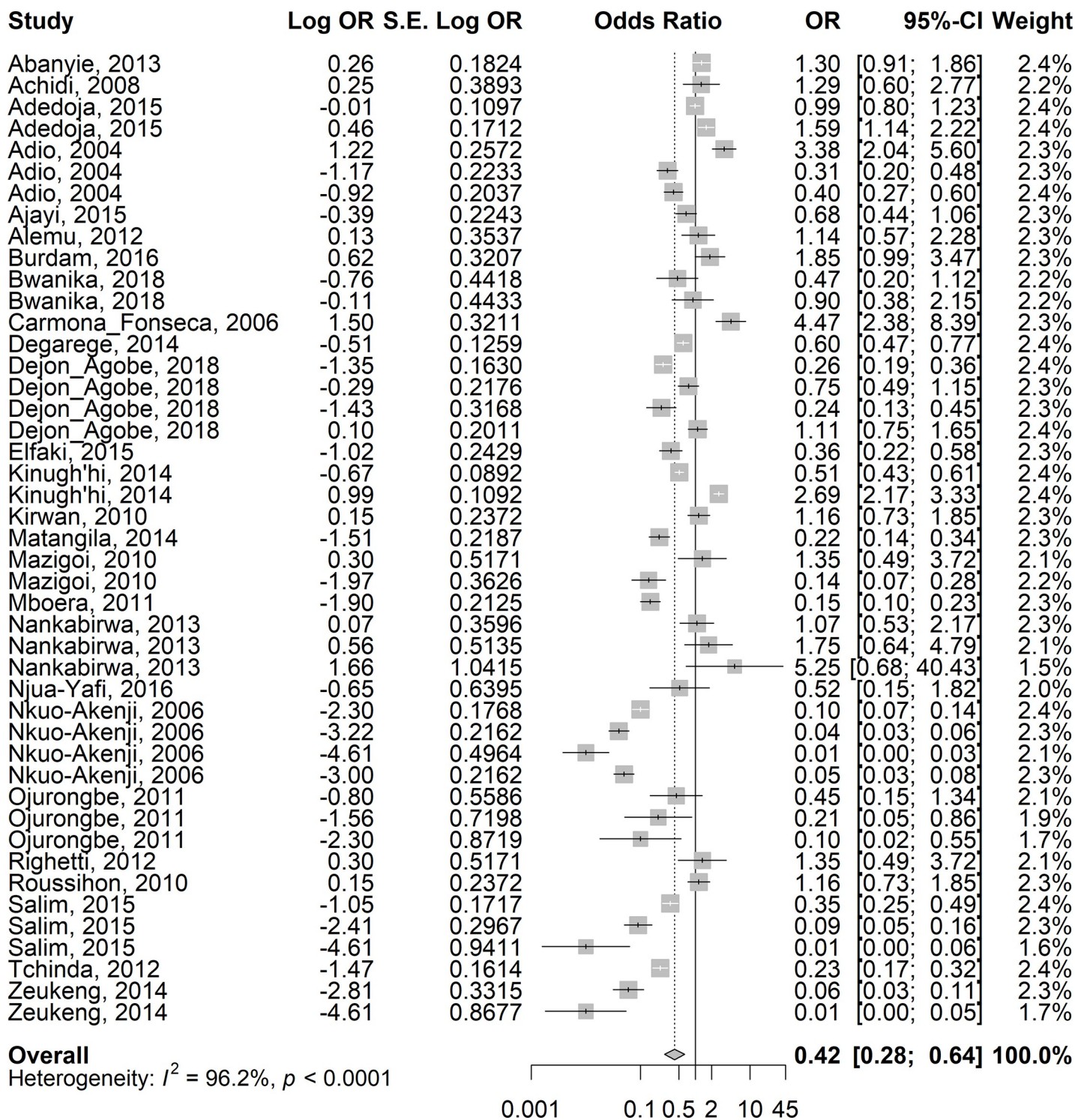

| Study | Log OR | S.E. Log OR | Odds Ratio | OR | 95%-CI | Weight |
|---|---|---|---|---|---|---|
| Abanyie, 2013 | 0.26 | 0.1824 | | 1.30 | [0.91; 1.86] | 2.4% |
| Achidi, 2008 | 0.25 | 0.3893 | | 1.29 | [0.60; 2.77] | 2.2% |
| Adedoja, 2015 | -0.01 | 0.1097 | | 0.99 | [0.80; 1.23] | 2.4% |
| Adedoja, 2015 | 0.46 | 0.1712 | | 1.59 | [1.14; 2.22] | 2.4% |
| Adio, 2004 | 1.22 | 0.2572 | | 3.38 | [2.04; 5.60] | 2.3% |
| Adio, 2004 | -1.17 | 0.2233 | | 0.31 | [0.20; 0.48] | 2.3% |
| Adio, 2004 | -0.92 | 0.2037 | | 0.40 | [0.27; 0.60] | 2.4% |
| Ajayi, 2015 | -0.39 | 0.2243 | | 0.68 | [0.44; 1.06] | 2.3% |
| Alemu, 2012 | 0.13 | 0.3537 | | 1.14 | [0.57; 2.28] | 2.3% |
| Burdam, 2016 | 0.62 | 0.3207 | | 1.85 | [0.99; 3.47] | 2.3% |
| Bwanika, 2018 | -0.76 | 0.4418 | | 0.47 | [0.20; 1.12] | 2.2% |
| Bwanika, 2018 | -0.11 | 0.4433 | | 0.90 | [0.38; 2.15] | 2.2% |
| Carmona_Fonseca, 2006 | 1.50 | 0.3211 | | 4.47 | [2.38; 8.39] | 2.3% |
| Degarege, 2014 | -0.51 | 0.1259 | | 0.60 | [0.47; 0.77] | 2.4% |
| Dejon_Agobe, 2018 | -1.35 | 0.1630 | | 0.26 | [0.19; 0.36] | 2.4% |
| Dejon_Agobe, 2018 | -0.29 | 0.2176 | | 0.75 | [0.49; 1.15] | 2.3% |
| Dejon_Agobe, 2018 | -1.43 | 0.3168 | | 0.24 | [0.13; 0.45] | 2.3% |
| Dejon_Agobe, 2018 | 0.10 | 0.2011 | | 1.11 | [0.75; 1.65] | 2.4% |
| Elfaki, 2015 | -1.02 | 0.2429 | | 0.36 | [0.22; 0.58] | 2.3% |
| Kinugh'hi, 2014 | -0.67 | 0.0892 | | 0.51 | [0.43; 0.61] | 2.4% |
| Kinugh'hi, 2014 | 0.99 | 0.1092 | | 2.69 | [2.17; 3.33] | 2.4% |
| Kirwan, 2010 | 0.15 | 0.2372 | | 1.16 | [0.73; 1.85] | 2.3% |
| Matangila, 2014 | -1.51 | 0.2187 | | 0.22 | [0.14; 0.34] | 2.3% |
| Mazigoi, 2010 | 0.30 | 0.5171 | | 1.35 | [0.49; 3.72] | 2.1% |
| Mazigoi, 2010 | -1.97 | 0.3626 | | 0.14 | [0.07; 0.28] | 2.2% |
| Mboera, 2011 | -1.90 | 0.2125 | | 0.15 | [0.10; 0.23] | 2.3% |
| Nankabirwa, 2013 | 0.07 | 0.3596 | | 1.07 | [0.53; 2.17] | 2.3% |
| Nankabirwa, 2013 | 0.56 | 0.5135 | | 1.75 | [0.64; 4.79] | 2.1% |
| Nankabirwa, 2013 | 1.66 | 1.0415 | | 5.25 | [0.68; 40.43] | 1.5% |
| Njua-Yafi, 2016 | -0.65 | 0.6395 | | 0.52 | [0.15; 1.82] | 2.0% |
| Nkuo-Akenji, 2006 | -2.30 | 0.1768 | | 0.10 | [0.07; 0.14] | 2.4% |
| Nkuo-Akenji, 2006 | -3.22 | 0.2162 | | 0.04 | [0.03; 0.06] | 2.3% |
| Nkuo-Akenji, 2006 | -4.61 | 0.4964 | | 0.01 | [0.00; 0.03] | 2.1% |
| Nkuo-Akenji, 2006 | -3.00 | 0.2162 | | 0.05 | [0.03; 0.08] | 2.3% |
| Ojurongbe, 2011 | -0.80 | 0.5586 | | 0.45 | [0.15; 1.34] | 2.1% |
| Ojurongbe, 2011 | -1.56 | 0.7198 | | 0.21 | [0.05; 0.86] | 1.9% |
| Ojurongbe, 2011 | -2.30 | 0.8719 | | 0.10 | [0.02; 0.55] | 1.7% |
| Righetti, 2012 | 0.30 | 0.5171 | | 1.35 | [0.49; 3.72] | 2.1% |
| Roussihon, 2010 | 0.15 | 0.2372 | | 1.16 | [0.73; 1.85] | 2.3% |
| Salim, 2015 | -1.05 | 0.1717 | | 0.35 | [0.25; 0.49] | 2.4% |
| Salim, 2015 | -2.41 | 0.2967 | | 0.09 | [0.05; 0.16] | 2.3% |
| Salim, 2015 | -4.61 | 0.9411 | | 0.01 | [0.00; 0.06] | 1.6% |
| Tchinda, 2012 | -1.47 | 0.1614 | | 0.23 | [0.17; 0.32] | 2.4% |
| Zeukeng, 2014 | -2.81 | 0.3315 | | 0.06 | [0.03; 0.11] | 2.3% |
| Zeukeng, 2014 | -4.61 | 0.8677 | | 0.01 | [0.00; 0.05] | 1.7% |
| **Overall** | | | | **0.42** | **[0.28; 0.64]** | **100.0%** |

Heterogeneity: $I^2$ = 96.2%, $p$ < 0.0001

**Fig 4. Forest plot showing the risk of asymptomatic/uncomplicated or severe *P. falciparum* infection in children co-infected with STH compared with children who did not have STH infection.**

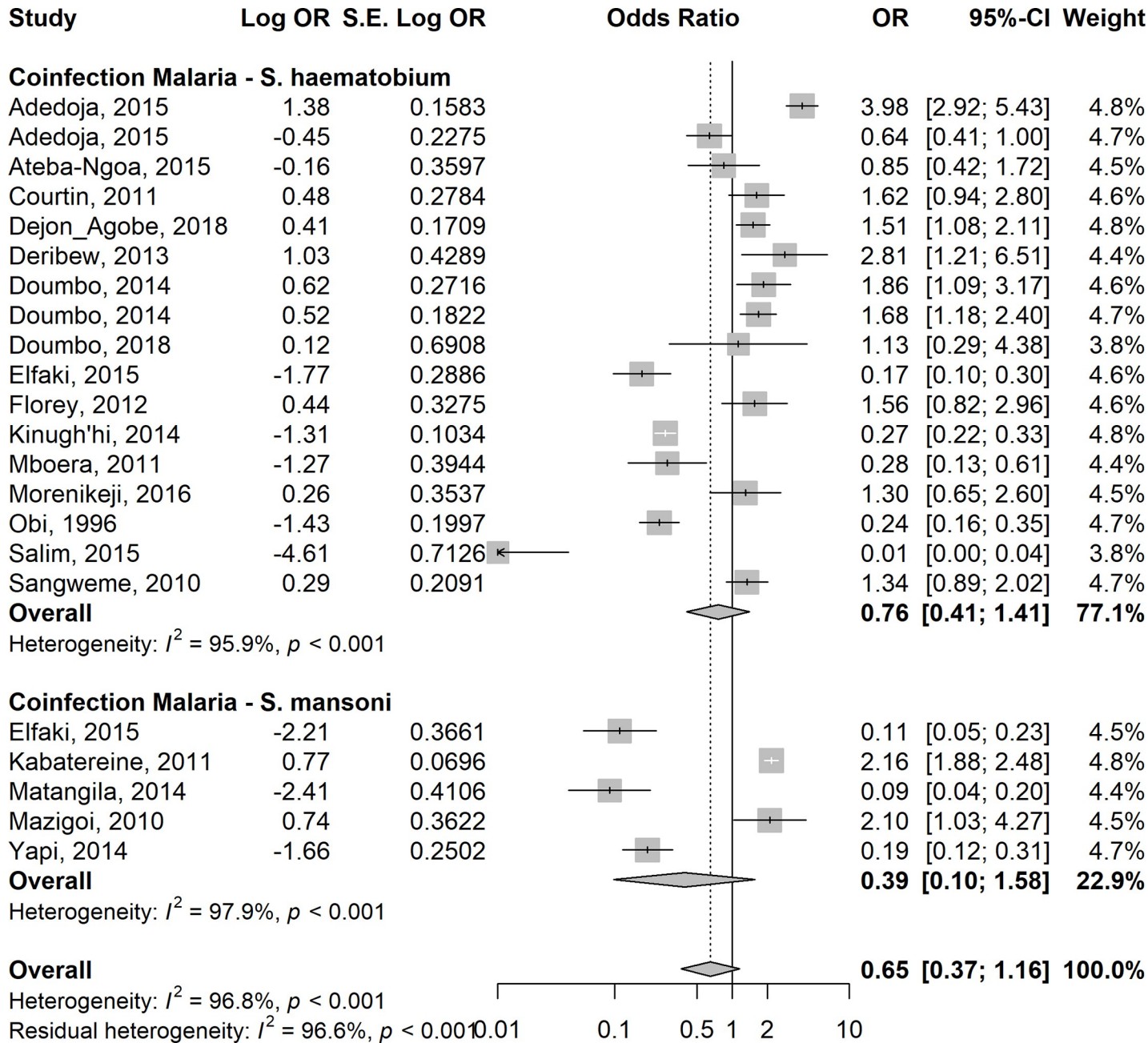

**Fig 5. Forest plot showing the risk of asymptomatic/uncomplicated or severe P. falciparum infection in children co-infected with *Schistosoma* spp compared with children who did not have *Schistosoma* infection.**

*Plasmodium*-helminth co-infections. The prevalence of hookworm-*Plasmodium* co-infection was significantly higher in boys than girls, but no gender difference was observed in the prevalence of *A. lumbricoides*-*Plasmodium* co-infection [42, 43]. Children had 9.3 times the odds of co-infection compared to adults (95%CI = 5.3–16.3), with children aged 7–9 years most frequently co-infected with *S. haematobium* followed by the age group 4–6 years [58]. This finding is in agreement with that of another study [52] but conflicts with other findings [85].

| Study | Log OR | S.E. Log OR | Odds Ratio | OR | 95%-CI | Weight |
|---|---|---|---|---|---|---|
| Abanyie, 2013 | -0.69 | 0.2892 | | 0.50 | [0.28; 0.88] | 6.3% |
| Adedoja, 2015 | -1.39 | 0.1546 | | 0.25 | [0.18; 0.34] | 6.4% |
| Adio, 2004 | 3.19 | 0.5524 | | 24.26 | [8.22; 71.63] | 5.7% |
| Ajayi, 2015 | -0.29 | 0.2586 | | 0.75 | [0.45; 1.24] | 6.3% |
| Alemu, 2012 | 2.62 | 0.5313 | | 13.80 | [4.87; 39.10] | 5.7% |
| Burdam, 2016 | 1.39 | 0.5327 | | 4.00 | [1.41; 11.36] | 5.7% |
| Humphries, 2011 | 0.69 | 1.0230 | | 1.99 | [0.27; 14.78] | 4.3% |
| Kinung'hi, 2014 | 0.52 | 0.0738 | | 1.69 | [1.46; 1.95] | 6.5% |
| Matangila, 2014 | -1.20 | 0.2256 | | 0.30 | [0.19; 0.47] | 6.4% |
| Mboera, 2011 | 0.58 | 0.4671 | | 1.78 | [0.71; 4.45] | 5.9% |
| Njua-Yafi, 2016 | -1.24 | 0.7184 | | 0.29 | [0.07; 1.19] | 5.2% |
| Njunda, 2015 | 1.79 | 0.1810 | | 5.99 | [4.20; 8.54] | 6.4% |
| Nkuo-Akenji, 2006 | -0.22 | 0.2673 | | 0.80 | [0.47; 1.35] | 6.3% |
| Nkuo-Akenji, 2006 | -2.12 | 0.2207 | | 0.12 | [0.08; 0.18] | 6.4% |
| Righetti, 2012 | -1.47 | 0.6792 | | 0.23 | [0.06; 0.87] | 5.3% |
| Tchinda, 2012 | 0.49 | 0.2632 | | 1.64 | [0.98; 2.75] | 6.3% |
| Zeukeng, 2014 | 0.97 | 0.7777 | | 2.63 | [0.57; 12.08] | 5.0% |
| **Overall** | | | | **1.20** | **[0.59; 2.45]** | **100.0%** |

Heterogeneity: $I^2$ = 96.3%, $p$ < 0.0001

**Fig 6. Forest plot showing odds of anaemia in children co-infected with *Plasmodium* species and pooled STH species compared with those who were infected with *P. falciparum* alone.**

**Interactions between malaria and helminth.** A Kenyan study [58] reported that the odds of intensity of *S. haematobium* infection increased with increasing *Plasmodium* infection intensity, consistent with similar with findings of studies from Senegal [48], Mali [67, 68],

| Study | Log OR | S.E. Log OR | Odds Ratio | OR | 95%-CI | Weight |
|---|---|---|---|---|---|---|
| Adedoja, 2015 | -0.80 | 0.1289 | | 0.45 | [0.35; 0.58] | 15.0% |
| Deribew, 2013 | 2.20 | 0.8498 | | 9.00 | [1.70; 47.60] | 11.5% |
| Kinung'hi, 2014 | -0.13 | 0.0717 | | 0.88 | [0.76; 1.01] | 15.0% |
| Kinung'hi, 2014 | -0.43 | 0.0745 | | 0.65 | [0.56; 0.75] | 15.0% |
| Matangila, 2014 | -2.66 | 0.4106 | | 0.07 | [0.03; 0.16] | 14.1% |
| Mboera, 2011 | 0.07 | 0.3177 | | 1.07 | [0.57; 1.99] | 14.4% |
| Sangweme, 2010 | 1.93 | 0.1503 | | 6.86 | [5.11; 9.21] | 14.9% |
| **Overall** | | | | **0.97** | **[0.30; 3.14]** | **100.0%** |

Heterogeneity: $I^2$ = 97.8%, $p$ < 0.0001

**Fig 7. Forest plot showing odds of anaemia in children co-infected with Plasmodium species and pooled Schistosoma spp compared with those who were infected with P. falciparum alone.**

Thailand [49] and Zimbabwe [17]. These findings conflict with those of two RCTs conducted in Madagascar which reported a negative interaction between *P. falciparum* and *S. haematobium* light infections [38], between *A. lumbricoides* and malaria parasite density [39] and of another study which showed that *S. mansoni* increased susceptibility to malaria [19]. Two studies [43, 47] reported that *A. lumbricoides* exerted a protective impact on the severity and patency of malaria clinical infections [50], in agreement with the findings of another study [56] which showed that administration of anti-helminth drugs lowered the prevalence and intensity of *A. lumbricoides* infections and also the prevalence of *Plasmodium* infection.

**Anaemia induced by malaria-helminth co-infection.** Seven [34, 35, 40, 44, 46, 53, 54] of the 27 studies analysed for narrative reviews also contributed to the data synthesis for the odds of anaemia in *Plasmodium*-helminth co-infection described above. Four of these studies [34, 35, 40, 46] showed a lower mean haemoglobin values for children with *Plasmodium*-helminth co-infections than children with either *Plasmodium* or helminth infection only. Humphries *et al* in two studies [35, 36] reported that the risk of anaemia was lower in Ghanaian children co-infected with malaria and hookworm than in those with malaria alone, suggesting that hookworm may modulate the pathogenesis of *P. falciparum*. This is similar with findings from Deribew *et al* [40], but deviates from findings of other studies [41, 46, 53, 54].

**Immuno-modulating effects of helminths on malaria.** Three studies [55, 57, 59] reported the immuno-modulating effect of helminths on the clinical course of malaria. A significant correlation was observed between the occurrence of malaria attacks, hookworm carriage and a decrease in cytophilic IgG1 and IgG3 malaria responses, suggesting that helminth infection might increase malaria morbidity through a Th2 worm-driven pattern of anti-malarial immune responses [55]. This is consistent with a Senegalese study [59] in which higher circulating levels of interleukin-10 in the plasma of co-infected children were associated with decreased anti-plasmodial IgG responses, but disagrees with the findings of another study in which no effect of *S. haematobium* was observed on the innate and adaptive immune response among Gabonese children infected with *P. falciparum* [57].

**Quality of the included studies.** We assessed the quality of the studies included in this review, and observed that majority of the studies had strong quality in design. Overall rating based on the assessment criteria showed that four studies were of strong quality, 41 studies were of moderate quality and the quality of 10 studies was weak (Table 1). None of the studies was excluded from the review as a result of quality issues.

## Discussion

### Burden of malaria-helminth co-infections

This systematic review with meta-analysis of 28 studies involving 22,114 children across 13 endemic countries showed an overall prevalence of malaria-helminth co-infection of 17.7%. Expectedly, sub-analysis showed that the prevalence varied according to geographic regions of LMIC, with the prevalence being highest in South America region and lowest in Southeast Asia. The small sample size of the single study which satisfied eligibility for inclusion in this review from South America, is likely to explain the very high prevalence observed in this region. This is in contrast with a low prevalence observed in Southeast Asia region, although the only included study had a relatively bigger sample size. The small number of studies reviewed from outside SSA means that no general conclusions can be made about the frequency of *Plasmodium*-helminth co-infections between continents. The pooled prevalence rates observed among the larger number of studies undertaken in SSA ranged from 13.1 to 30.7%. These are similar findings to those of a previous review [86]. Apart from environmental factors which promote the co-existence of malaria and helminth infections in LMIC, a high

prevalence of co-infection in these settings has been attributed to prevalent infrastructural and behavioural problems such as poor sanitation, lack of toilet facilities, unsafe drinking water, and ineffective public health enlightenment programmes [86].

## Nature of interactions between malaria and helminths

Meta-analysis of the 24 studies in this review revealed lower odds of prevalent malaria in children co-infected with either STH or *Schistosoma* spp, but higher odds in children with malaria-STH co-infection when adjusted for confounders. This is similar to the findings of reviews conducted by Dagarege *et al* [7]. Also, narrative reviews conducted as part of this systematic review showed conflicting findings, with positive linear relationship between *Plasmodium falciparum* and *S.haematobium* in Kenyan, Senegalese, Thai and Zimbabwean children [17, 48, 49, 58] but negative interactions reported in Malagasy children [38, 39] and between *P. falciparum and S.mansoni* in another Senegalese paediatric population [19]. These conflicting findings underscore the complexity in the interactions between *P.falciparum* and helminths. The variations may also be influenced by the nature of immune responses elicited by the dual infections. Given the variability in the mechanisms of immune activation by helminths and *Plasmodium* parasites [16], and the fact that helminths downregulate immune responses to *Plasmodium* pathogens [57, 87], previous studies suggested that STH infections increase risk to *Plasmodium* infection and related clinical outcomes [55, 65].

## Risk of anaemia from malaria-helminth co-infections

The summary estimates from 16 studies in this review showed that the odds of anaemia were higher in children who were co-infected with *Plasmodium* and STH species than in children infection with *Plasmodium* alone. These findings are consistent with review by Naing *et al* [26], but not with findings of review by Dagarege *et al* [7]. The negative effect of *Plasmodium*-helminth co-infection on anaemia is not surprising considering the different mechanisms by which anaemia is produced in the two infections [2, 63, 88,89].

The pooled estimates from six studies in this review of *Plasmodium-Schistosoma* co-infection showed an equal odd of anaemia in co-infected children compared with uninfected children. This is not in keeping with the finding of a previous review [8], which reported a higher mean haemoglobin concentration in children co-infected with *S. haematobium* and *P. falciparum* than in those infected with only *P. falciparum*. A reason for this difference might be due to the relatively low number of primary studies used in calculating the summary estimates for the mean haemoglobin differences. Nevertheless, the finding of this review supports the postulated mechanisms modulating anaemia in *Schistosoma-Plasmodium* co-infection [40, 55, 59, 74].

## Study limitations

This systematic review did not escape from the limitations identified in previous reviews. Most of the primary studies included in this reviews were cross-sectional in design, making it challenging to conclusively establish the prevalence of malaria-helminth co-infections in LMIC. A very high level of heterogeneity was also observed among the included studies, although Egger's test did not reach statistical significance thereby ruling out the possibility of publication bias. Bias may have been introduced given that the primary studies were conducted in widely diverse populations, with significant variations in the study designs and methodologies adopted in implementing these primary studies. Nevertheless, this review has updated the body of knowledge on malaria-helminth co-infection through a comprehensive synthesis of data obtained from studies conducted across LMIC.

## Conclusion

In conclusion, we have shown that the prevalence of malaria-helminth co-infection is high in studies which evaluated the prevalence of both groups of infections in children living in endemic countries. The nature of interactions between malaria and helminth infection and impact of the co-infection on anaemia remain inconclusive, and may be modulated by immune responses of the affected children. Findings of this review support our resolve to employ improved tools to quantify the burden of malaria-helminth co-infections. This may generate better understanding of burden of the co-infection, which could be deployed for planning and implementation of appropriate interventions for integrated control and, ultimately elimination of both groups of infections. Exploring the dynamics of immune regulation in malaria and helminth co-infection may also be useful in malaria vaccine development to understand the influence of the dual infection on vaccine immunogenicity [2].

## Supporting information

**S1 Fig. Forest plot showing pooled prevalence of malaria-helminth co-infection by geographic region.**
(DOCX)

**S2 Fig.** a-b-c: Forest plot showing sub-group analysis of summary estimates of prevalence of *P.falciparum-A. lumbricoides* co-infection, forest plot showing sub-group analysis of summary estimates of prevalence of *P.falciparum*-hookworm co-infection, forest plot showing sub-group analysis of summary estimates of prevalence of *P.falciparum-T.trichuria* co-infection.
(DOCX)

**S3 Fig.** a-b: Forest plot showing sub-group analysis of prevalence of *Plasmodium-S. haematobium* co-infection in children in LMIC, forest plot showing sub-group analysis of prevalence of *Plasmodium-S.mansoni* co-infection in children in LMIC.
(DOCX)

**S4 Fig. Forest plot showing the risk of asymptomatic/uncomplicated or severe P. falciparum infection in children co-infected with STH compared with children who did not have STH infection when adjusted for geographic location of the study participants.**
(DOCX)

**S5 Fig. Forest plot showing the risk of asymptomatic/uncomplicated or severe P. falciparum infection in children co-infected with STH compared with children who did not have STH infection when adjusted for age, gender, socio-economic status and nutrition status of the children.**
(DOCX)

**S1 File. Study protocol including the search strategy.**
(PDF)

## Acknowledgments

We appreciate the support and guidance on the refinement of the search strategy by Jane Falconer and colleagues at the Library and Archive Services of the London School of Hygiene & Tropical Medicine, UK.

## Author Contributions

**Conceptualization:** Muhammed O. Afolabi.

**Data curation:** Muhammed O. Afolabi, Boni M. Ale, Edgard D. Dabira, Schadrac C. Agbla, Brian Greenwood.

**Formal analysis:** Muhammed O. Afolabi, Boni M. Ale.

**Funding acquisition:** Muhammed O. Afolabi, Brian Greenwood.

**Investigation:** Muhammed O. Afolabi, Edgard D. Dabira, Jean Louis A. Ndiaye.

**Methodology:** Muhammed O. Afolabi.

**Project administration:** Muhammed O. Afolabi, Brian Greenwood.

**Resources:** Brian Greenwood.

**Supervision:** Muhammed O. Afolabi, Amaya L. Bustinduy, Brian Greenwood.

**Writing – original draft:** Muhammed O. Afolabi.

**Writing – review & editing:** Muhammed O. Afolabi, Boni M. Ale, Edgard D. Dabira, Schadrac C. Agbla, Amaya L. Bustinduy, Jean Louis A. Ndiaye, Brian Greenwood.

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
