## [Decision Letter · Decision Letter 0]

1 Dec 2020

Dear Dr Afolabi,

Thank you very much for submitting your manuscript "Malaria and helminth co-infections in children living in low- and middle-income countries: a systematic review with meta-analysis" for consideration at PLOS Neglected Tropical Diseases. As with all papers reviewed by the journal, your manuscript was reviewed by members of the editorial board and by several independent reviewers. The reviewers appreciated the attention to an important topic. Based on the reviews, we are likely to accept this manuscript for publication, providing that you modify the manuscript according to the review recommendations. 

Sincerely,

Jessica K Fairley, MD, MPH

Associate Editor

Kristien Verdonck

Deputy Editor

Reviewer's Responses to Questions

**Key Review Criteria Required for Acceptance?**

**Methods**

-Are the objectives of the study clearly articulated with a clear testable hypothesis stated?

-Is the study design appropriate to address the stated objectives?

-Is the population clearly described and appropriate for the hypothesis being tested?

-Is the sample size sufficient to ensure adequate power to address the hypothesis being tested?

-Were correct statistical analysis used to support conclusions?

-Are there concerns about ethical or regulatory requirements being met?

Reviewer #1: -The objectives of the study are clearly articulated.

-The study design is appropriate to address the stated objectives.

-The population is clearly described and appropriate for the hypothesis.

-The sample size has has added to the body of knowledge byt further studies are recommended.

-Correct statistical analysis were used to support conclusions.

-Are there no concerns about ethical or regulatory requirements

Reviewer #2: Abstract

 Background

Line # 159-162

 The authors need to describe the study objective clearly. It is a mixed of objective and utility of this review. 

Methodology/Principal Findings

1) Line # 44-47 Information provided is not yet clear enough 

Please, rephrase these sentences. . The authors chose a random model due to a substantial heterogeneity among studies (i.e. I2 test more than?? %). Please, provide a relevant ref (e.g. Higgins et al 2019. Cochrane Handbook for Systematic Reviews of Interventions version 6 ) 

2) To be more informative, please provide I 2 values (after OR and its 95%I) for important outcomes.

3) Line # 58……………..It will be better to describe “A subset of 16 studies …..” 

Text

Eligibility Criteria 

1) Line # 172-76

Because of observational studies, PECO format should be more applicable. 

It will be better to present the inclusion criteria addressing each acronym 

 For example, 

Participants ( P): children aged 1-16 years living in LMIC, 

2) Line # 177

Adults or pregnant women were excluded. To be more informative, please give justifications in the light of immune status or pathophysiology. 

3) Line # 204-205

It will be better to provide kappa statistic. 

Data Analysis

Line # 230

 If adjusted OR was not available in an included study, how would you treat such study?

 The authors did not provide how they had estimated a pooled prevalence. (On line # 293, the authors provided the results of pooled prevalence. If so, the method of pooling prevalence should have been described in the method section). For more details, please, see a ref (Nyaga VN, Arbyn M, Aerts M. Metaprop: a Stata command to perform meta-analysis of binomial data. Arch Public Health. 2014;72(1):39). This paper addressed tests of significance on the pooled proportion which typically rely on normal probabilities. It also, address Freeman-Tukey double arcsine transformation (this ref paper is aimed to address metaprop package in STATA. But, it is applicable for methodological aspect of pooled prevalence, regardless of software used.)

Reviewer #3: The objectives of the study are clearly articulated with a clear testable hypothesis.

The study design is appropriate to address the research question

The population description is not appropriate for the hypothesis being tested, since the pathogens study or reviewed are not limited to the LMIC, rather to endemic regions. The age groups are not well define and seems to be limited to infants without explaining the rationnal to exclude adults or not.

The title of the review including LMIC, but there is no rationale under this specification. Would co-infection of helminth and malaria differ in non-LMIC? One could simply consider in endemic areas

The keys word includes paediatrics population, which is too specific. Better consider pre- and school children 

The sample size calculation is not applicable in the review.

The statistical analysis is appropriate to support the conclusion.

The ethical issue is not applicable. However the review is registerred.

**Results**

-Does the analysis presented match the analysis plan?

-Are the results clearly and completely presented?

-Are the figures (Tables, Images) of sufficient quality for clarity?

Reviewer #1: The analysis presented match the analysis plan and the results are clearly and completely presented.

However the data in Figure 2 presents P-Schisto first and P-STH afterwards which in the reverse to the data presented in the Abstract and the main text. I recommend reversing the data presentin Figure 2 to match the flow of the text.

The age range(s) in Table 1 for manuscripts 9, 10, 11, 23 and 44 are unclear.

Reviewer #2: The current presentation is not yet smooth. It will be better to improve the flow of presentation. 

Having mentioned above, it will be better to provide I2 values.

Reviewer #3: The analysis presented match the analysis plan

In general the results are clearly presented, howver, two sub title overlap. line 334 Anaemia in Plasmodium- helminth co-infections" and line 365: Anaemia induced by malaria-helminth co-infection" . More distinction are needed.

The other reviews on co-infection should be reviewed and taking in consideration to avoid the redundant since some facts are known already

**Conclusions**

-Are the conclusions supported by the data presented?

-Are the limitations of analysis clearly described?

-Do the authors discuss how these data can be helpful to advance our understanding of the topic under study?

-Is public health relevance addressed?

Reviewer #1: The conclusions are supported by the data presented.

The limitations of the analysis (suitable published manuscripts from LMIC) are clearly described.

The authors discuss how these data can be helpful to advance our understanding of the topic under study and the need fro further studies and analysis.

The public health relevance of immune-regulation should be further addressed and potentially its long-term implications if programs continue to make progress towards control/elimination.

Reviewer #2: It is difficult to catch the salient points. It will be better to present with the use of subheadings, including study limitations.

Reviewer #3: The conclusion are clear and supported by data presented. The limitation are described, as well as public health relevance.

**Editorial and Data Presentation Modifications?**

Reviewer #1: Minor Revision to address suggestions to Figure 2 and Table 1 and expand the discussion on immune regulation.

Reviewer #2: (No Response)

Reviewer #3: (No Response)

**Summary and General Comments**

Reviewer #1: This is a well researched and well presented paper. 

The topic is highly significant as the global burden of STH and Schito reduce and the complicated interaction with malaria, anaemia and possibly other infections such as Covid-19 are so little understood.

Reviewer #2: It is fine.

Reviewer #3: The review entitled " Malaria and helminth co-infections in children living in low- and middle-income

countries: a systematic review with meta-analysis "undertake, by Dr. Affolabi et al, is useful and mark a new step regarding the rationale research question of co-infection of malaria and helminths in term of susceptibility and immune responses. However some points need to be addressed ahead.

The title of the review including LMIC, but there is no rationale under this specification. Would co-infection of helminth and malaria differ non-LMIC? One could simply consider in endemic areas

The keys word includes paediatrics population, which is too specific. Better consider pre- and school children 

On line 85, the egg detection methods are too limited. Please elaborate a bite more

All studies announced are not refferenced

Two sub title overlap. line 334 Anaemia in Plasmodium- helminth co-infections" and line 365: Anaemia induced by malaria-helminth co-infection" 

Other reviews on co-infection should be reviewed to avoid the redundant since some facts are known already

PLOS authors have the option to publish the peer review history of their article (what does this mean?). If published, this will include your full peer review and any attached files.

Reviewer #1: No

Reviewer #2: No

Reviewer #3: No
---

## [Editor Report · Decision Letter 1]

13 Jan 2021

Dear Dr Afolabi,

We are pleased to inform you that your manuscript 'Malaria and helminth co-infections in children living in endemic countries: a systematic review with meta-analysis' has been provisionally accepted for publication in PLOS Neglected Tropical Diseases.

Best regards,

Jessica K Fairley, MD, MPH

Associate Editor

Kristien Verdonck

Deputy Editor

---

## [Editor Report · Acceptance letter]

4 Feb 2021

Dear Dr Afolabi,

We are delighted to inform you that your manuscript, "Malaria and helminth co-infections in children living in endemic countries: a systematic review with meta-analysis," has been formally accepted for publication in PLOS Neglected Tropical Diseases.

Best regards,

Shaden Kamhawi

co-Editor-in-Chief

Paul Brindley

co-Editor-in-Chief
